# Can AI Generate Love Advice?: Toward Neural Answer Generation for Non-Factoid Questions

**Makoto Nakatsuji, Hisashi Ito, Naruhiro Ikeda, Shota Sagara & Akihisa Fujita**
NTT Resonant Inc.

{nakatuji,h-ito,nikeda,s-sagara,akihisa}@nttr.co.jp

## Abstract

Deep learning methods that extract answers for non-factoid questions from QA sites are seen as critical since they can assist users in reaching their next decisions through conversations with AI systems. The current methods, however, have the following two problems: (1) They can not understand the ambiguous use of words in the questions as word usage can strongly depend on the context (e.g. the word "relationship" has quite different meanings in the categories of Love advice and other categories). As a result, the accuracies of their answer selections are not good enough. (2) The current methods can only *select* from among the answers held by QA sites and can not *generate* new ones. Thus, they can not answer the questions that are somewhat different with those stored in QA sites. Our solution, Neural Answer Construction Model, tackles these problems as it: (1) Incorporates the biases of semantics behind questions (e.g. categories assigned to questions) into word embeddings while also computing them regardless of the semantics. As a result, it can extract answers that suit the contexts of words used in the question as well as following the common usage of words across semantics. This improves the accuracy of answer selection. (2) Uses biLSTM to compute the embeddings of questions as well as those of the sentences often used to form answers (e.g. sentences representing conclusions or those supplementing the conclusions). It then simultaneously learns the optimum combination of those sentences as well as the closeness between the question and those sentences. As a result, our model can construct an answer that corresponds to the situation that underlies the question; it fills the gap between answer selection and generation and is the first model to move beyond the current simple answer selection model for non-factoid QAs. Evaluations using datasets created for love advice stored in the Japanese QA site, Oshiete goo, indicate that our model achieves 20 % higher accuracy in answer creation than the strong baselines. Our model is practical and has already been applied to the love advice service in Oshiete goo.

## 1 Introduction

Recently, dialog-based natural language understanding systems such as Apple's Siri, IBM's Watson, Amazon's Echo, and Wolfram Alpha have spread through the market. In those systems, Question Answering (QA) modules are particularly important since people want to know many things in their daily lives. Technically, there are two types of questions in QA systems: factoid questions and non-factoid ones. The former are asking, for instance, for the name of a person or a location such that "What/Who is $X$?". The latter are more diverse questions which cannot be answered by a short fact. They range from advice on making long distance relationships work well, to requests for opinions on some public issues. Significant progress has been made at answering factoid questions (Wang et al. (2007); Yu et al. (2014)), however, retrieving answers for non-factoid questions from the Web remains a critical challenge in improving QA modules. The QA community sites such as Yahoo! Answers and Quora can be sources of training data for the non-factoid questions where the goal is to automatically select the best of the stored candidate answers.

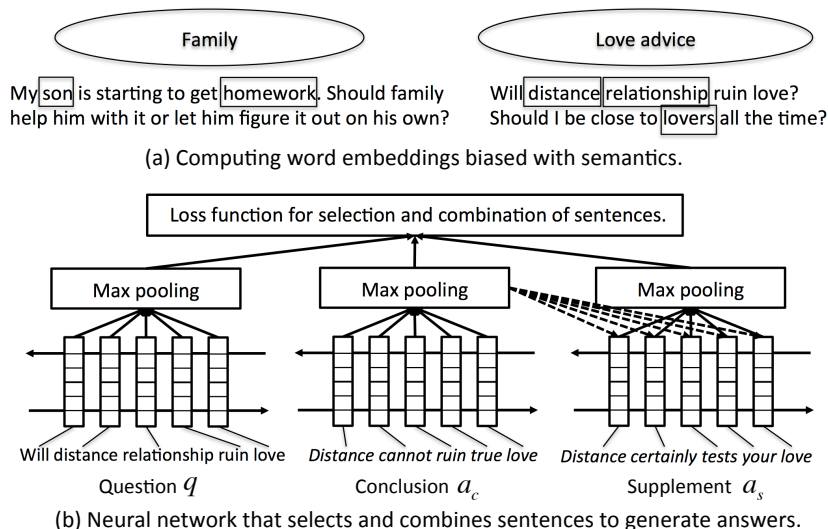

(a) Computing word embeddings biased with semantics.

(b) Neural network that selects and combines sentences to generate answers.

Figure 1: Main ideas: (a) word embeddings with semantics and (b) a neural answer construction.

Recent deep learning methods have been applied to this non-factoid answer selection task using datasets stored in the QA sites resulting in state-of-the-art performance (Yu et al. (2014); Tan et al. (2015); Qiu & Huang (2015); Feng et al. (2015); Wang & Nyberg (2015); Tan et al. (2016)). They usually compute closeness between questions and answers by the individual embeddings obtained using a convolutional model. For example, Tan et al. (2016) builds the embeddings of questions and those of answers based on bidirectional long short-term memory (biLSTM) models, and measures their closeness by cosine similarity. It also utilizes an efficient attention mechanism to generate the answer representation following the question context. Their results show that their model can achieve much more accurate results than the strong baseline (Feng et al. (2015)). The current methods, however, have the following two problems when applying them to real applications:

(1) They can not understand the ambiguous use of words written in the questions as words are used in quite different ways following the context in which they appear (e.g. the word "relationship" used in a question submitted to "Love advice" category is quite different from the same word submitted to "Business advice" category). This makes words important for a specific context likely to be disregarded in the following answer selection process. As a result, the answer selection accuracies become weak for real applications.

(2) They can only *select* from among the answers stored in the QA systems and can not *generate* new ones. Thus, they can not answer the questions that are somewhat different from those stored in the QA systems even though it is important to cope with such differences when answering non-factoid questions (e.g. questions in the "Love advice" category are often different due to the situation and user even though they share the same topics.). Furthermore, the answers selected from QA datasets often contain a large amount of unrelated information. Some other studies have tried to create short answers to the short questions often seen in chat systems (Vinyals & Le (2015); Serban et al. (2015)). Our target, non-factoid questions in QA systems, are, however, much longer and more complicated than those in chat systems. As described in their papers, the above methods, unfortunately, create unsatisfying answers to such non-factoid questions.

To solve the above problems, this paper proposes a neural answer construction model; it fills the gap between answer selection and generation and is the first model to move beyond the current simple answer selection model for non-factoid QAs. It extends the above mentioned biLSTM model since it is language independent and free from feature engineering, linguistic tools, or external resources. Our model takes the following two ideas:

(1) Before learning answer creation, it incorporates semantic biases behind questions (e.g. titles or categories assigned to questions) into word vectors while computing vectors by using QA documents stored across semantics. This process emphasizes the words that are important for a certain context. As a result, it can select the answers that suit the contexts of words used in the questions as well as the common usage of words seen across semantics. This improves the accuracies of answer selections.

For example, in Fig. 1-(a), there are two questions in category "Family" and "Love advice". Words marked with rectangles are category specific (i.e. "son" and "homework" are specifically observed in "Family" while "distance", "relationship", and "lovers" are found in "Love advice".) Our method can emphasize those words. As a result, answers that include the topics, "son" and "homework", or topics, "distance", "relationship", and "lovers", will be scored highly for the above questions in the following answer selection task.

(2) The QA module designer first defines the abstract scenario of answer to be created; types of sentences that should compose the answer and their occurrence order in the answer (e.g. typical answers in "Love advice" are composed in the order of the sentence types "sympathy", "conclusion", "supplementary for conclusion", and "encouragement"). The sentence candidates can be extracted from the whole answers by applying sentence extraction methods or sentence type classifiers (Schmidt et al. (2014); Zhang et al. (2008); Nishikawa et al. (2010); Chen et al. (2010)). It next simultaneously learns *the closeness* between questions and sentences that may include answers as well as *combinational optimization* of those sentences. Our method also uses an attention mechanism to generate sentence representations according to the prior sentence; this extracts important topics in the sentence and tracks those topics in subsequent sentences. As a result, it can construct answers that have natural sentence flow whose topics correspond to the questions. Fig. 1-(b) explains the proposed neural-network by using examples. Here, the QA module designer first defines the abstract scenario for the answer as in the order of "conclusion" and "supplement". Thus, there are three types of inputs "question", "conclusion", and "supplement". It next runs biLSTMs over those inputs separately; it learns the order of word vectors such that "relationships" often appears next to "distance". It then computes the embedding for the question, that for conclusion, and that for supplement by max-pooling over the hidden vectors output by biLSTMs. Finally, it computes the closeness between question and conclusion, that between question and supplement, and combinational optimization between conclusion and supplement with the attention mechanism, simultaneously (dotted lines in Fig. 1-(b) represent attention from conclusion to supplement).

We evaluated our method using datasets stored in the Japanese QA site Oshiete goo[1]. In particular, our evaluations focus on questions stored in the "Love advice" category since they are representative non-factoid questions: the questions are often complicated and most questions are very long. The results show that our method outperforms the previous methods including the method by (Tan et al. (2016)); our method accurately constructs answers by naturally combining key sentences that are highly close to the question.

## 2 RELATED WORK

Previous works on answer selection normally require feature engineering, linguistic tools, or external resources. Recent deep learning methods are attractive since they demonstrate superior performance compared to traditional machine learning methods without the above mentioned tiresome procedures. For example, (Wang & Nyberg (2015); Hu et al. (2014)) construct a joint feature vector on both question and answer and then convert the task into a classification or ranking problem. (Feng et al. (2015); Yu et al. (2014); dos Santos et al. (2015); Qiu & Huang (2015)) learn the question and answer representations and then match them by certain similarity metrics. Recently, Tan et al. (2016) took the latter approach and achieved more accurate results than the current strong baselines (Feng et al. (2015); Bendersky et al. (2011)). They, however, can only select answers and not generate them. Other than the above, recent neural text generation methods (Serban et al. (2015); Vinyals & Le (2015)) can also intrinsically be used for answer generation. Their evaluations showed that they could generate very short answer for factoid questions, but not the longer and more complicated answers demanded by non-factoid questions. Our Neural Answer Construction Model fills the gap between answer selection and generation for non-factoid QAs. It simultaneously learns *the closeness* between questions and sentences that may include answers as well as *combinational optimization* of those sentences. Since the sentences themselves in the answer are short, they can be generated by neural conversation models like (Vinyals & Le (2015));

As for word embeddings with semantics, some previous methods use the semantics behind words by using semantic lexicons such as WordNet and Freebase (Xu et al. (2014); Bollegala et al. (2016); Faruqui et al. (2015); Johansson & Nieto Piña (2015)). They, however, do not use the semantics be-

---
[1] http://oshiete.goo.ne.jp

hind the question/answer documents; e.g. document categories. Thus, they can not well catch the contexts in which the words appear in the QA documents. They also require external semantic resources other than QA datasets.

## 3 PRELIMINARY

Here, we explain QA-LSTM (Tan et al. (2015)), the basic discriminative framework for answer selection based on LSTM, since we base our ideas on its framework.

We first explain the LSTM and introduce the terminologies used in this paper. Given input sequence $\mathbf{X} = \{\mathbf{x}(1), \mathbf{x}(2), \cdots, \mathbf{x}(N)\}$, where $\mathbf{x}(t)$ is $t$-th word vector, $t$-th hidden vector $\mathbf{h}(t)$ is updated as:

$$
\begin{aligned}
\mathbf{i}_t &= \sigma(\mathbf{W}_i\mathbf{x}(t) + \mathbf{U}_i\mathbf{h}(t-1) + \mathbf{b}_i) \\
\mathbf{f}_t &= \sigma(\mathbf{W}_f\mathbf{x}(t) + \mathbf{U}_f\mathbf{h}(t-1) + \mathbf{b}_f) \\
\mathbf{o}_t &= \sigma(\mathbf{W}_o\mathbf{x}(t) + \mathbf{U}_o\mathbf{h}(t-1) + \mathbf{b}_o) \\
\widetilde{\mathbf{c}}_t &= \tanh(\mathbf{W}_c\mathbf{x}(t) + \mathbf{U}_c\mathbf{h}(t-1) + \mathbf{b}_c) \\
\mathbf{c}_t &= \mathbf{i}_t * \widetilde{\mathbf{c}}_t + \mathbf{f}_t * \mathbf{c}_{t-1} \\
\mathbf{h}(t) &= \mathbf{o}_t * \tanh(\mathbf{c}_t)
\end{aligned}
$$

There are three gates (input $\mathbf{i}_t$, forget $\mathbf{f}_t$, and output $\mathbf{o}_t$), and a cell memory vector $\mathbf{c}_t$. $\sigma$ is the sigmoid function. $\mathbf{W} \in R^{H \times N}$, $\mathbf{U} \in R^{H \times H}$, and $\mathbf{b} \in R^{H \times 1}$ are the network parameters to be learned. Single-direction LSTMs are weak in that they fail to make use of the contextual information from the future tokens. BiLSTMs use both the previous and future context by processing the sequence in two directions, and generate two sequences of output vectors. The output for each token is the concatenation of the two vectors from both directions, i.e. $\overleftrightarrow{h(t)} = \overrightarrow{h(t)} \parallel \overleftarrow{h(t)}$.

In the QA-LSTM framework, given input pair $(q, a)$ where $q$ is a question and $a$ is a candidate answer, it first retrieves the word embeddings (WEs) of both $q$ and $a$. Next, it separately applies a biLSTM over the two sequences of WEs. Then, it generates fixed-sized distributed vector representations $\mathbf{o}_q$ for $q$ (or $\mathbf{o}_a$ for $a$) by computing max pooling over all the output vectors and then concatenating the resulting vectors on both directions of the biLSTM. Finally, it uses cosine similarity $\cos(\mathbf{o}_q, \mathbf{o}_a)$ to score the input $(q, a)$ pair.

It then defines the training objective as the hinge loss of:

$$
\mathcal{L} = \max\{0, M - \cos(\mathbf{o}_q, \mathbf{o}_a^+) + \cos(\mathbf{o}_q, \mathbf{o}_a^-)\}
$$

where $\mathbf{o}_a^+$ is an output vector for ground truth answer, $\mathbf{o}_a^-$ is that for an incorrect answer randomly chosen from the entire answer space, and $M$ is a margin. It treats any question with more than one ground truth as multiple training examples. Finally, batch normalization is performed on the representations before computing cosine similarity (Ioffe & Szegedy (2015)).

## 4 METHOD

We first explain our word embeddings with semantics.

### 4.1 WORD EMBEDDINGS WITH DOCUMENT SEMANTICS

This process is inspired by paragraph2vec (Le & Mikolov (2014)); an unsupervised algorithm that learns fixed-length feature representations from variable-length pieces of texts, such as sentences, paragraphs, and documents.

First, we explain paragraph2vec model. It averages the paragraph vector with several word vectors from a paragraph and predicts the following word in the given context. It trains both word vectors and paragraph vectors by stochastic gradient descent and backpropagation (Rumelhart et al. (1988)). While paragraph vectors are unique among paragraphs, the word vectors are shared.

Next, we introduce our method that incorporates the semantics behind QA documents into word embeddings (WEs) in the training phase. The idea is simple. Please see Fig. 2. It averages the vector

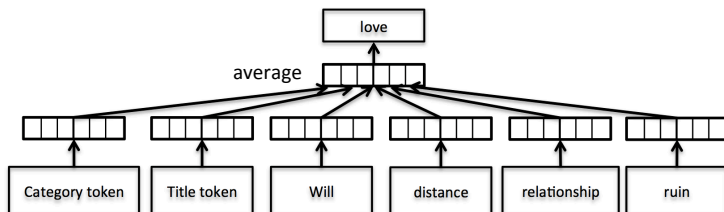

Figure 2: Learning word vectors biased with semantics.

of category token and the vectors of title tokens, which are assigned to the QA documents, with several of the word vectors present in those documents. It then predicts the following word in the given context. Here, title tokens are defined by nouns that are extracted from titles assigned to the question. Multiple title tokens can be extracted from a title while one category token is assigned to a question. Those tokens are shared among datasets in the same category. It trains the category vector and title vectors as well as word vectors in QA documents as per paragraph2vec model. Those additional vectors are used as semantic biases for learning WEs. They are useful in emphasizing the words following the contexts of particular categories or titles. This improves the accuracies of answer selection described later as explained in Introduction.

For example, in Fig. 2, it can incorporate semantic biases from category "Love advice" into the words (e.g. "Will", "distance", "relationship", "ruin", "love" and so on) in the question in "Love advice". Thus, it can well apply the biases from category "Love advice" to the words (e.g. "distance" and "relationship") if they specifically appear in "Love advice". On the other hand, words that appear in several categories (e.g. "will") are biased with several categories and thus will not be emphasized.

## 4.2 NEURAL ANSWER CONSTRUCTION MODEL

Here, we explain our model. We first explain our approach and then the algorithm.

**Approach**  It takes the following three approaches:

- **Design the abstract scenario for the answer:** The answer is constructed according to the order of the sentence types defined by the designer. For example, there are the sentence types such as sentence that states sympathy with the question, sentence that states a conclusion to the question, sentence that supplements the conclusion, and sentence that states encouragement to the questioner. This is inspired by the automated web service composition framework (Rao & Su (2005)) where the requester should build an abstract process before the web service composition planning starts. In our setting, the process is the scenario of answer and the service is the sentence in the scenario. Thus, our method can construct an answer by binding concrete sentences to fit the scenario.

  For example, the scenario for love advice can be designed as follows: it begins with a sympathy sentence (e.g. "You are struggling too."), next it states a conclusion sentence (e.g. "I think you should make a declaration of love to her as soon as possible."), then it supplements the conclusion by a supplemental sentence (e.g. "If you are too late, she maybe fall in love with someone else."), and finally it ends with an encouragement sentence (e.g. "Good Luck!").

- **Joint neural network to learn sentence selection and combination:** Our model computes *the combination optimization* among sentences that may include the answer as well as *the closeness* between question and sentences within a single neural network. This improves answer sentence selection; our model can avoid the cases in which the combination of sentences are not good enough though the scores of closeness between the question and each sentence are high. It also can let the parameter tuning simpler than the model that separates the network for sentence selection and that for sentence combination. The image of this neural-network is depicted in Fig. 1-(b). Here, it learns the closeness between sentence "Will distance relationship ruin love?" and "Distance cannot ruin true love", the closeness between "Will distance relationship ruin love?" and "Distance certainly tests your love.", and the combination between "Distance cannot ruin true love' and "Distance certainly tests your love.".

---

**Algorithm 1** A neural answer construction model

---

**Input:** Pairs of question, conclusion, and supplement, $\{(q, a_c, \text{and } a_s)\}$.
**Output:** Parameters set by the algorithm.
 1: **for** $n = 1$, $n{+}{+}$, while $n < N$ **do**
 2: **for** each pair $(q, a_c, a_s)$ **do**
 3: Computes $\mathbf{o}_q^c$ and $\mathbf{o}_c$ by biLSTMs and max pooling.
 4: Computes $\mathbf{o}_q^s$ by biLSTM and max pooling.
 5: **for** each $t$-th hidden vector for supplement **do**
 6: Computes $\tilde{\mathbf{h}}_s(t)$ by Eq. (1).
 7: **end for**
 8: Computes $\mathbf{o}_s$ by max pooling.
 9: Computes $\mathcal{L}$ by Eq. (2).
10: **end for**
11: **end for**

---

- **Attention mechanism to improve the combination of sentences :** Our method extracts important topics in the conclusion sentence and emphasizes those topics in the supplemental sentence in the training phase; this is inspired by (Tan et al. (2016)) who utilizes an attention mechanism to generate the answer representation following the question context. As a result, it can combine conclusions with the supplements following the contexts written in the conclusion sentences. This makes the story in the created answers very natural. In Fig. 1-(b), our attention mechanism extracts important topics (e.g. topic that represents "distance") in the conclusion sentence "Distance cannot ruin true love" and emphasizes those topics in computing the representation of the supplement sentence "Distance certainly tests your love.".

**Procedure** The core part of the answer is usually the conclusion sentence and its supplemental sentence. Thus, for simplicity, we here explain the procedure of our model in selecting and combining the above two types of sentences. As the reader can imagine, it can easily be applied to four sentence types. Actually, our love advice service by AI in oshiete-goo was implemented for four types of sentences, sympathy, conclusion, supplement, and encouragement (see Evaluation section). The model is illustrated in Fig. 1-(b) in which the input pair is $(q, a_c, a_s)$ where $q$ is the question, $a_c$ is a candidate conclusion sentence, and $a_s$ is a candidate supplemental sentence. The word embeddings (WEs) for words in $q$, $a_c$, and $a_s$ are extracted in the way described in the previous subsection. The procedure of our model is as follows (please see the Algorithm 1 also.):

(1) It iterates the following procedures (2) to (7) $N$ times (line 1 in the algorithm).

(2) It picks up each pair $(q, a_c, \text{and } a_s)$ in the dataset (line 2 in the algorithm).

In the following steps (3) and (4), the same biLSTM is applied to both $q$ and $a_c$ to compute the closeness between $q$ and $a_c$. Similarly, the same biLSTM is applied to both $q$ and $a_s$. However, the biLSTM for computing closeness between $q$ and $a_c$ differs from that between $q$ and $a_s$ since $a_c$ and $a_s$ have different characteristics.

(3) It separately applies a biLSTM over the two sequences of WEs, $q$ and $a_c$, and computes the max pooling over the $t$-th hidden vector for question $\mathbf{h}_q^c(t)$ and that for conclusion $\mathbf{h}_c(t)$. As a result, it acquires the question embedding, $\mathbf{o}_q^c$ and the conclusion embedding, $\mathbf{o}_c$ (line 3 in the algorithm).

(4) It also separately applies a biLSTM over the two sequences of WEs, $q$ and $a_s$, and computes the max pooling over the $t$-th hidden vector for question $\mathbf{h}_q^s(t)$ to acquire the question embedding, $\mathbf{o}_q^s$ (line 4 in the algorithm). $\mathbf{o}_q^s$ is different from $\mathbf{o}_q^c$ since our method does not share the sub-network used for computing closeness between $q$ and $a_c$ and that between $q$ and $a_s$ as described above.

(5) It applies the attention mechanism from conclusion to supplement. Specifically, given the output vector of biLSTM on the supplemental side at time step $t$, $\mathbf{h}_s(t)$, and the conclusion embedding, $\mathbf{o}_c$, the updated vector $\tilde{\mathbf{h}}_s(t)$ for each conclusion token is formulated as below (line 6 in the algorithm):

Table 1: Comparison of AP for answer selection.

|        | QA-LSTM | Attentive-LSTM | Semantic-LSTM | Construction | Our method |
|--------|---------|----------------|---------------|--------------|------------|
| $K=1$  | 0.8472  | 0.8196         | 0.8499        | 0.8816       | *0.8846*   |
| $K=3$  | 0.8649  | 0.844566       | 0.8734        | 0.8884       | 0.8909     |
| $K=5$  | 0.8653  | 0.8418         | 0.8712        | 0.8827       | 0.8845     |
| $K=10$ | 0.8603  | 0.8358         | 0.8658        | 0.8618       | 0.8647     |

Table 2: Comparison of AP for answer construction.

|        | QA-LSTM | Attentive-LSTM | Semantic-LSTM | Construction | Our method |
|--------|---------|----------------|---------------|--------------|------------|
| $K=1$  | 0.3262  | 0.3235         | 0.3664        | 0.3813       | *0.3901*   |
| $K=3$  | 0.3753  | 0.3694         | 0.4078        | 0.5278       | 0.5308     |
| $K=5$  | 0.3813  | 0.3758         | 0.4133        | 0.5196       | 0.5271     |
| $K=10$ | 0.3827  | 0.3777         | 0.4151        | 0.4838       | 0.4763     |

$$
\begin{aligned}
\mathbf{m}_{s,c}(t) &= \tanh(\mathbf{W}_{sm}\mathbf{h}_s(t) + \mathbf{W}_{cm}\mathbf{o}_c) \\
\mathbf{s}_{s,c}(t) &= \exp(\mathbf{w}_{mb}{}^{\mathrm{T}}\mathbf{m}_{s,c}(t)) \\
\tilde{\mathbf{h}}_s(t) &= \mathbf{h}_s(t)\mathbf{s}_{s,c}(t)
\end{aligned}
\tag{1}
$$

$\mathbf{W}_{sm}$, $\mathbf{W}_{cm}$, and $\mathbf{w}_{mb}$ are attention parameters. Conceptually, the attention mechanism gives more weights on words that include important topics in the conclusion sentence.

(6) It computes the max pooling over $\tilde{\mathbf{h}}_s(t)$ and acquires the supplemental embedding, $\mathbf{o}_s$ (line 8 in the algorithm).

(7) It computes the closeness between question and conclusion and that between question and supplement as well as the optimization combination between conclusion and supplement. The training objective is given as (line 9 in the algorithm):

$$
\begin{aligned}
\mathcal{L} &= \max\{0, M-(\cos(\mathbf{o}_q, [\mathbf{o}_c^+, \mathbf{o}_s^+])-\cos(\mathbf{o}_q, [\mathbf{o}_c^+, \mathbf{o}_s^-]))\} \\
&+ \max\{0, M-(\cos(\mathbf{o}_q, [\mathbf{o}_c^+, \mathbf{o}_s^+])-\cos(\mathbf{o}_q, [\mathbf{o}_c^-, \mathbf{o}_s^+]))\} \\
&+ \max\{0, (1+k)M-(\cos(\mathbf{o}_q, [\mathbf{o}_c^+, \mathbf{o}_s^+])-\cos(\mathbf{o}_q, [\mathbf{o}_c^-, \mathbf{o}_s^-]))\} \\
&+ \max\{0, M-(\cos(\mathbf{o}_q, [\mathbf{o}_c^+, \mathbf{o}_s^-])-\cos(\mathbf{o}_q, [\mathbf{o}_c^-, \mathbf{o}_s^-]))\} \\
&+ \max\{0, M-(\cos(\mathbf{o}_q, [\mathbf{o}_c^-, \mathbf{o}_s^+])-\cos(\mathbf{o}_q, [\mathbf{o}_c^-, \mathbf{o}_s^-]))\}
\end{aligned}
\tag{2}
$$

where $[\mathbf{y}, \mathbf{z}]$ is the concatenation of two vectors, $\mathbf{y}$ and $\mathbf{z}$, $\mathbf{o}_q$ is $[\mathbf{o}_q^c, \mathbf{o}_q^s]$, $\mathbf{o}^+$ is an output vector for a ground truth answer, and $\mathbf{o}^-$ is that for an incorrect answer randomly chosen from the entire answer space. In the above equation, the first (or second) term presents the loss that occurs when both question-conclusion pair (q-c) and question-supplemental pair (q-s) are correct while q-c (or q-s) is correct but q-s (or q-c) is incorrect. The third term computes the loss that occurs when both q-c and q-s are correct while both q-c and q-s are incorrect. The fourth (or fifth) term computes the loss that occurs when q-c (or q-s) is correct but q-s (or q-c) is incorrect while both q-c and q-s are incorrect. $M$ is constant margin and $k$ ($0 < k < 1$) is a parameter controlling the margin. Thus, the resulting margin for the third term is larger than those for other terms. In this way, by considering the case when either conclutions or supplements are incorrect or not, this equation optimizes the combinations among conclusion and supplement. In addition, it can take the closeness between question and conclusion (or supplement) in consideration by cosine similarity.

The parameter sets $\{\mathbf{W}_i, \mathbf{W}_f, \mathbf{W}_o, \mathbf{W}_c, \mathbf{U}_i, \mathbf{U}_f, \mathbf{U}_o, \mathbf{U}_c, \mathbf{b}_i, \mathbf{b}_f, \mathbf{b}_o, \mathbf{b}_c\}_c$ for question-conclusion matching, $\{\mathbf{W}_i, \mathbf{W}_f, \mathbf{W}_o, \mathbf{W}_c, \mathbf{U}_i, \mathbf{U}_f, \mathbf{U}_o, \mathbf{U}_c, \mathbf{b}_i, \mathbf{b}_f, \mathbf{b}_o, \mathbf{b}_c\}_s$ for question-supplement matching, and $\{\mathbf{W}_{sm}, \mathbf{W}_{cm}, \mathbf{w}_{mb}\}$ for conclusion-supplement attention are trained during the iterations. After the model is trained, our method uses $\cos(\mathbf{o}_q, [\mathbf{o}_c, \mathbf{o}_s])$ to score the input ($q$, $a_c$, $a_s$) pair and constructs an answer that has a conclusion and its supplement.

## 5 EVALUATION

We used our method to select or construct answers to the questions stored in "Love advice" category.

Table 3: Comparison of human evaluation results.

| QA-LSTM | (1) | (2) | (3) | (4) | Our method | (1) | (2) | (3) | (4) |
|---------|-----|-----|-----|-----|-----------|-----|-----|-----|-----|
|         | 57  | 66  | 41  | 36  |           | *116* | 51 | 13 | 20 |

Table 4: Examples of answers created by *QA-LSTM* and those by *Our method*.

| Questions | Answers created by *QA-LSTM* | Answers created by *Our method* |
|-----------|------------------------------|--------------------------------|
| I'm afraid to confess my love to her, what should I do? | You should wait until you feel excited. If you interact with her indifferently, it will be difficult to develop any relation with her. | It is better to concentrate on how to confess your love to her. I understand you are struggling since you love her very much. |
| A guy I like says to me "I like you at home" kiddingly. It may be the ordinary gentleness. Some hope? | You don't have to test his love immediately. Unless he likes you, he would not have gone to see a movie with you. | Yes, there is some hope. You can understand his reaction more easily if your understanding of each other is deeper. |
| I seldom meet an interesting person. I worry about how to become close to him. Should I approach to him positively? | Try to select your words correctly. Unless you confess your love to him, it is difficult to convey your emotion to him. | You should confess your love to him. Unless you confess your love to him, it is difficult to convey your emotion to him. |

## 5.1 DATASET

We evaluated our method using a dataset stored in Japanese online QA service Oshiete-goo. First, the word embeddings were built by using 189,511 questions and their 771,956 answers stored in 16 categories including "Love Advice", "Traveling", and "Health Care". 6,250 title tokens were extracted from the titles. Then, we evaluated answer selection and construction tasks by using a corpus containing about 5,000 question-conclusion-supplement sentences. Conclusions and supplement sentences were extracted by human experts from answers. The readers could use sentence extraction methods (Schmidt et al. (2014); Zhang et al. (2008); Nishikawa et al. (2010); Chen et al. (2010)) or neural conversation models like (Vinyals & Le (2015)) to semi-automatically extract/generate those sentences.

## 5.2 COMPARED METHODS

We compared the accuracy of the following five methods:

- *QA-LSTM* proposed by (Tan et al. (2015)).
- *Attentive LSTM*: introduces an attention mechanism from question to answer and is evaluated as the current best answer selection method Tan et al. (2016).
- *Semantic LSTM*: performs answer selection by using our word embeddings biased with semantics.
- *Construction*: performs our proposed answer construction without attention mechanism.
- *Our method*: performs our answer construction with attention mechanism from conclusion to supplement.

## 5.3 METHODOLOGY AND PARAMETER SETUP

We randomly divided the dataset into two halves, training dataset and predicted one, and conducted two-fold cross validation. Results shown later are the average values.

Both for answer selection and construction, we used Average Precision (AP) against the top-K ranked answers in the results because we consider that the most highly ranked answers are important for users. If the number of ranked items is $K$, the number of correct answers among the top-j ranked items $N_j$, and the number of all correct answers (paired with the questions) $D$, AP is defined as follows:

$$AP = \frac{1}{D} \sum_{1 \leq j \leq K} \frac{N_j}{j}$$

For answer construction, we checked whether each method could recreate the original answers. As the reader easily can understand, this is a much more difficult task than answer selection and thus the values of AP will be smaller than the results for answer selection.

We tried word vectors and qa vectors of different sizes, and finally set the word vector size to 300 and the LSTM output vectors for biLSTMs to $50 \times 2$. We also tried different margins in the hinge

loss function, and fixed the margin, $M$, to 0.2 and $k$ to 1.0. The iteration count $N$ was set to 20. For our method, the embeddings for questions, those for conclusions, and those for supplements were pretrained by *Semantic LSTM* before answer construction since this enhances the overall accuracy.

We did not use attention mechanism from question to answer for *Semantic LSTM*, *Construction* and *Our method*. This is because, as we present in the results subsection, the lengths of questions are much longer than those of answer sentences, and thus the attention mechanism from question to answer became noise for sentence selection.

## 5.4 RESULTS

We now present the results of the evaluations.

**Answer Selection**   We first compare the accuracy of methods for answer selection. The results are shown in Table 1. *QA-LSTM* and *Attentive LSTM* are worse than *Semantic-LSTM*. This indicates that *Semantic-LSTM* can incorporate semantic information (titles/categories) into word embeddings; it can emphasize words according to the context they appeared and thus the matching accuracy between question vector and conclusion (supplement) vector was improved. *Attentive LSTM* is worse than *QA-LSTM* as described above. *Construction* and *Our method* are better than *Semantic-LSTM*. This is because they can avoid the combinations of sentences that are not good enough even though the scores of closeness between questions and sentences are high. This implies that, if the combination is not good, the selection of answer sentences also tends to be erroneous. Finally, *Our method*, which provides sophisticated selection/combination strategies, yielded higher accuracy than the other methods. It achieved 4.4% higher accuracy than *QA-LSTM* (*QA-LSTM* marked 0.8472 while *Our method* marked 0.8846.).

**Answer Construction**   We then compared the accuracy of the methods for answer construction. Especially for the answer construction task, the top-1 result is most important since many QA applications show only the top-1 answer. The results are shown in Table 2. There is no answer construction mechanism in *QA-LSTM*, *Attentive-LSTM*, and *Semantic-LSTM*. Thus we simply merge the conclusion and supplement, each of which has the highest similarity with the question by each method. *QA-LSTM* and *Attentive LSTM* are much worse than *Semantic-LSTM*. This is because the sentences output by *Semantic-LSTM* are selected by utilizing the words that are emphasized for a context for "Love advice" (i.e. category and titles). *Construction* is better than *Semantic-LSTM* since it simultaneously learns the optimum combination of sentences as well as the closeness between the question and sentences. Finally, *Our method* is better than *Construction*. This is because it well employs the attention mechanism to link conclusion and supplement sentences and thus the combinations of the sentences are more natural than those of *Construction*. *Our method* achieved 20% higher accuracy than *QA-LSTM* (*QA-LSTM* marked 0.3262 while *Our method* marked 0.3901.).

The computation time for *our method* was less than two hours. All experiments were performed on NVIDIA TITAN X/Tesla M40 GPUs, and all methods were implemented by Python in the Chainer framework. Thus, our method well suits real applications. In fact, it is already being used in the love advice service of Oshiete goo [2].

**Human evaluation**   The outputs of *QA-LSTM* and *Our method* were judged by two human experts. The experts entered the questions, which were not included in our evaluation datasets, to the AI system and rated the created answers based on the following scale: (1) the conclusion and supplement sentences as well as their combination were good, (2) the sentences were good in isolation but their combination was not good, (3) One of the selections (conclusion or supplement) was good but their combination was not good, and (4) both sentences and their combination were not good. The answers were judged as good if they satisfied the following two points: (A) the contents of answer sentences correspond to the question. (B) the story between conclusion and supplement is natural.

The results are shown in Table 3. Table 4 presents examples of the questions and answers constructed (they were originally Japanese and translated into English for readability. The questions are summarized since the original ones were very long.). The readers can also see Japanese answers from our service URL presented the above. Those results indicate that the experts were much more satisfied with the outputs of *Our method* than those by *QA-LSTM*; 58 % of the answers created by *Our method* were classified as (1). This is because, as can be see in Table 4, *Our method* can naturally

---

[2]http://oshiete.goo.ne.jp/ai

combine the sentences as well as select sentences that match the question. It well coped with the questions that were somewhat different from those stored in the evaluation dataset.

Actually, when the public used our love advice service, it was surprising to find that the 455 answers created by the AI whose name is oshi-el (uses *Our method*) were judged as *Good answers* by users from among the 1,492 questions entered from September 6th to November 5th[3]. The rate of getting Good answers by oshi-el is twice that of the average human user in oshiete-goo when we focus on users who answered more than 100 questions in love advice category. Thus, we think this is a good result.

## 6    CONCLUSION

This is the first study that create answers for non-factoid questions. Our method incorporates the biases of semantics behind questions into word embeddings to improve the accuracy of answer selection. It then simultaneously learns the optimum combination of answer sentences as well as the closeness between questions and sentences. Our evaluation shows that our method achieves 20 % higher accuracy in answer construction than the method based on the current best answer selection method. Our model presents an important direction for future studies on answer generation. Since the sentences themselves in the answer are short, they can be generated by neural conversation models like (Vinyals & Le (2015)); this means that our model can be extended to generate complete answers once the abstract scenario is made.

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
