# Peer review of "CAN AI GENERATE LOVE ADVICE?: TOWARD NEURAL ANSWER GENERATION FOR NON-FACTOID QUESTIONS"

_ICLR 2017 — rejected_

[Reviewer Comment · AnonReviewer3 · rating 4 · 17 Dec 2016]
**No Title**

This paper extends mostly on top of the work of QA-biLSTM and QA-biLSTM with attentions, as proposed in Tan et al. 2015 and Tan et al. 2016, in the following 2 ways:

1. It trains a topic-specific word embedding using an approach similar to Paragraph2vec by leveraging the topic and title information provided in the data.

2. It considers the multiple-unit answer selection problem (e.g., one sentence selected from answer section, and another selected from supplemental section) vs. the single answer selection problem as studied in Tan et al 2015 and 2016. The mechanism used to retain the coherence between different parts of the answers is inspired by the attention mechanism introduced by Tan et al. 2016.

While the practical results presented in the paper is interesting, the main innovations of this paper are rather limited.

[Official Review · AnonReviewer2 · rating 4 · confidence 4 · 17 Dec 2016]

This paper proposes a neural architecture for answering non-factoid questions. The author's model improves over previous neural models for answer sentence selection. Experiments are conducted on a Japanese love advice corpus; the coolest part of the paper for me was that the model was actually rolled out to the public and its answers were rated twice as good as actual human contributors! 

It was hard for me to determine the novelty of the contribution. The authors mention that their model "fills the gap
between answer selection and generation"; however, no generation is actually performed by the model! Instead, the model appears to be very similar to the QA-LSTM of Tan et al., 2015 except that there are additional terms in the objective to handle conclusion and supplementary sentences. The structure of the answer is fixed to a predefined template (e.g., conclusion --> supplementary), so the model is not really learning how to order the sentences. The other contribution is the "word embedding with semantics" portion described in sec 4.1, which is essentially just the paragraph vector model except with "titles" and "categories" instead of paragraphs. 

While the result of the paper is a model that has actually demonstrated real-life usefulness, the technical contributions do not strike me as novel enough for publication at ICLR.

Other comments:
- One major issue with the reliance of the model on the template is that you can't evaluate on commonly-used non-factoid QA datasets such as InsuranceQA. If the template were not fixed beforehand (but possibly learned by the model), you could conceivably evaluate on different datasets. 
- The examples in Table 4 don't show a clear edge in answer quality to your model; QA-LSTM seems to choose good answers as well.
- Doesn't the construction model have an advantage over the vanilla QA-LSTM in that it knows which sentences are conclusions and which are supplementary? Or does QA-LSTM also get this distinction?

[Official Review · AnonReviewer1 · rating 4 · confidence 4 · 20 Dec 2016]
**Abstract patterns do not seem to be generalizable across different types of non-factoid questions and need more analysis of results**

Summary: The paper presents an approach – Neural Answer Construction Model for the task of answering non-factoid questions, in particular, love-advice questions. The two main features of the proposed model are the following – 1) it incorporates the biases of semantics behind questions into word embeddings, 2) in addition to optimizing for closeness between questions and answers, it also optimizes for optimum combination of sentences in the predicted answer. The proposed model is evaluated using the dataset from a Japanese online QA service and is shown to outperform the baseline model (Tan et al. 2015) by 20% relatively (6% absolutely). The paper also experiments with few other baseline models (ablations of the proposed model).

Strengths:

1. The two motivations behind the proposed approach – need to understand the ambiguous use of words depending on context, and need to generate new answers rather than just selecting from answers held by QA sites – are reasonable.

2. The novelty in the paper involves the following – 1) incorporating biases of semantics behind questions into word embeddings using paragraph2vec like model, modified to take as inputs - words from questions, question title token and question category token, 2) modelling optimum combination of sentences (conclusion and supplement sentences) in the predicted answer, 3) designing abstract scenario for answers, inspired by automated web-service composition framework (Rao & Su (2005)), and 4) extracting important topics in conclusion sentence and emphasizing them in supplemental sentence using attention mechanism (attention mechanism is similar to Tan et al. 2016).

3. The proposed method is shown to outperform the current best method (Tan et al. 2015) by 20% relatively (6% absolutely) which seems to be significant improvement. 

4. The paper presents few ablations studies that provide insights on how much different components of the model (such as incorporating biases into word embeddings, incorporating attention from conclusion to supplement) are helping towards performance improvement.

Weaknesses/Suggestions/Questions:

1. How are the abstract patterns determined, i,e., how did the authors determine that the answers to love-advice questions generally constitute of sympathy, conclusion, supplement for conclusion and encouragement? How much is the improvement in performance when using abstract patterns compared to the case when not using these patters, i.e. when candidate answers are picked from union of all corpuses rather than picking from respective corpuses (corpuses for sympathy, conclusion etc.).

2. It seems that the abstract patterns are specific to the type of questions. So, the abstract patterns for love-advice will be different from those for business advice. Thus, it seems like the abstract patterns need to be hand-coded for different types and hence one model cannot generalize across different types.

3. The paper should present explicit analysis of how much combinational optimization between sentences help – comparison between model performance with and without combinational optimization keeping rest of the model architecture same. The authors could also plot the accuracy of the model as a function of the combinational optimization scores. This will provide insights into how significant are the combinational optimization scores towards overall model accuracy.


4. Paper says that current systems designed for non-factoid QA cannot generalize to questions outside those stored in QA sites and claims that this is one of the contributions of this paper. In order to ground that claim, the paper should show experimentally how well the proposed method generalized to such out-of-domain questions. Although the questions asked by human experts in the human evaluation were not from the evaluation datasets, the paper should analyze how different those questions were compared to the questions present in the evaluation datasets.

5. For human evaluation, were the outputs of the proposed model and that of the QA-LSTM model judged each judged by both the human experts OR one of the human experts judged the outputs of one system and the other human expert judged the outputs of the other system? If both the sets of outputs were each judged by both the human experts, how were the ratings of the two experts combined for every questions? 

6. I wonder why the authors did not do a human evaluation where they just ask human workers (not experts) to compare the output of the proposed model with that of the QA-LSTM model – which of the two outputs they would like to hear when asking for advice. Such an evaluation would not get biased by whether each sentence is good or not, whether the combination is good or not. Looking at the qualitative examples in Table 4, I personally like the output of the QA-LSTM more than that of the proposed model because they seem to provide a direct answer to the question (e.g., for the first example the output of the QA-LSTM says “You should wait until you feel excited”, whereas the output of the proposed model says “It is better to concentrate on how to confess your love to her” which seems a bit indirect to the question asked.)

7. Given a question, is the ground-truth answer different in the two tasks -- answer selection and answer construction?

8. The paper mentions that Attentive LSTM (Tan et al. 2016) is evaluated as the current best answer selection method (section 5.2). So, why is its accuracy lower than that of QA-LSTM in table 1. The authors explain this by pointing out the issue of questions being very long compared to answers and hence the attention being noisy. But, did these issues not exist in the dataset used by Tan et al. 2016?

9. The paper says the proposed method achieves 20% gain over current best (in Conclusion section) where they refer to QA-LSTM as the current best method. However, in the description of Attentive LSTM (section 5.2), the paper mentions that Attention LSTM is the current best method. So, could authors please clarify the discrepancy?

10. Minor correction: remove space between 20 and % in abstract.

Review Summary: The problem of non-factoid QA being dealt with in the paper is an interesting and useful problem to solve. The motivations presented in the paper behind the proposed approach are reasonable. The experiments show that the proposed model outperforms the baseline model. However, the use of abstract patterns to determine the answer seems like hand-designing and hence it seems like these abstract patterns need to be designed for every other type of non-factoid question and hence the proposed approach is not generalizable to other types. Also, the paper needs more analysis of the results to provide insights into the contribution of different model components.

[Final Decision · Program Chairs · 06 Feb 2017]
**ICLR committee final decision**

The program committee appreciates the authors' response to clarifying questions. Unfortunately, all reviewers are leaning against accepting the paper. Authors are encouraged to incorporate reviewer feedback in future iterations of this work.